# Bandit-Driven Batch Selection
# for Robust Learning under Label Noise

**Michal Lisicki**
University of Guelph, Vector Institute
`mlisicki@uoguelph.ca`

**Mihai Nica**
University of Guelph, Vector Institute
`nicam@uoguelph.ca`

**Graham W. Taylor**
University of Guelph, Vector Institute
`gwtaylor@uoguelph.ca`

## Abstract

We introduce a novel approach for batch selection in Stochastic Gradient Descent (SGD) training, leveraging combinatorial bandit algorithms. Our methodology focuses on optimizing the learning process in the presence of label noise, a prevalent issue in real-world datasets. Experimental evaluations on the CIFAR-10 dataset reveal that our approach consistently outperforms existing methods across various levels of label corruption. Importantly, we achieve this superior performance without incurring the computational overhead commonly associated with auxiliary neural network models. This work presents a balanced trade-off between computational efficiency and model efficacy, offering a scalable solution for complex machine learning applications.

## 1 Introduction

As applications increasingly demand larger and more complex deep learning models, the need for efficient training strategies has become paramount. One way to accelerate training and potentially improve model performance is through the use of Curriculum Learning (CL) and adaptive batch selection. These techniques optimize learning by selectively focusing on data samples that are intrinsically rich and informative at the most appropriate stages of the learning process. Such strategies not only accelerate convergence but also enhance the model's ability to generalize [19, 20, 27].

While many methods use difficulty metrics to select easy, hard, or uncertain instances for training [31], a key area lies in handling noisy or mislabeled datasets [28]. This domain is particularly important for two reasons: a) the impact of batch selection strategies is easily measured, leading to more insightful conclusions; and b) it addresses the prevalent real-world scenarios where data is often sourced from the web [18] or crowdsourced [7], and a large portion is considered "unclean".

Sample selection strategies using auxiliary Deep Neural Networks (DNN) effectively mitigate the impact of noisy or mislabeled data. However, these approaches incur substantial computational overhead, limiting their scalability [9, 13, 17, 33]. While alternative methods like SELFIE [26] offer computational efficiency, they are under-explored and rely on steps like re-labeling for optimal performance. Meanwhile, the literature on CL and batch selection offers numerous methods for efficient sample selection across diverse domains [8, 19].

This paper introduces a novel approach that synergizes insights from the CL and batch selection literature to enhance efficient sampling schemes, specifically targeting scenarios with prevalent label noise. Our methodology aims to achieve superior performance without the computational burden

Workshop on Advancing Neural Network Training (full) and Workshop on Optimization in Machine Learning (abbreviated) at 37th Conference on Neural Information Processing Systems (NeurIPS 2023).

often associated with deploying additional DNNs, thereby striking a balance between efficacy and computational efficiency. Unlike traditional CL approaches that focus on individual instances or tasks, our method refines the feedback loop from each training iteration to optimize the *selected batch*. This approach is particularly relevant for tackling the challenges posed by the increasing computational complexity and diversity of machine learning applications across various domains.

## 2 Background

**Batch selection and curriculum learning**   CL [3] and its variants like Self-Paced Learning (SPL) [15] and Hard-Example Mining (HEM) [6, 19] provide frameworks for adaptive instance, batch, or task selection based on difficulty or importance. Despite the efficacy of these strategies in enhancing stability and convergence, a universal solution remains elusive, prompting exploration into varied strategies, new importance metrics, and advanced re-weighting and sampling techniques.

Re-weighting the model's loss by instance, akin to importance-sampling techniques, has been investigated by [6, 19, 24]. These studies have indicated that re-weighting can stabilize gradient estimates and reduce bias in the original objective function. However, both Loshchilov and Hutter [19] and Chang et al. [6] have argued that the impact of this strategy on performance is limited, and that comparable or superior results can be achieved by sampling from a weight-induced distribution. Matiisen et al. [20] compared the *sample selection* strategies, $\varepsilon$-greedy, Boltzmann, and Thompson sampling, and concluded that the optimal strategy hinges heavily on the sample weight metric. A novel metric not tied to difficulty was introduced by Chang et al. [6], emphasizing samples with high prediction uncertainty, and inspired by active learning. The authors demonstrated that, by avoiding overly easy or hard instances, their strategy surpassed SPL or HEM on datasets like MNIST and CIFAR with and without label noise. Song et al. [27] introduced Recency Bias to boost SGD's convergence by combining principles of [19] and [6]. The technique centers on prediction uncertainty, measured by predicted label entropy, as its adaptive sample selection metric. It uses a Boltzmann distribution with energy based on prediction uncertainty and a pressure parameter. Leveraging a sliding window, it emphasizes recent scores, mitigating overfitting and slow convergence. Importantly, Active Bias, Recency Bias, and our approach add minimal computational load to model training.

Automated curriculum learning (ACL) distinguishes itself among CL approaches by the degree of control over the learning process. In ACL, the selection of tasks is determined dynamically using an algorithm, typically an RL or a bandit method. Graves et al. [8] and Matiisen et al. [20] have proposed utilizing a non-stationary bandit (Exp3). They demonstrated that when an agent lacks prior knowledge of its tasks, ACL can significantly boost training efficiency relative to uniform sampling. Moreover, a bandit algorithm can discover complex orderings and opportunities for efficient knowledge transfer in an unsorted curriculum. Although prior literature has focused primarily on task-based ACL, the same principles can be utilized in instance and batch selection.

This paper aims to build upon the mentioned foundational CL techniques by introducing efficient batch selection methods, particularly in the context of learning with noisy labels.

**Efficient learning with noisy labels**   Learning with Noisy Labels (LNL) shares a connection with batch selection but has a different objective. While batch selection picks instances that inherently aid training, LNL focuses on distinguishing between those with clean and noisy labels. Both fields converge when the right batch selection strategy is used to isolate the "clean" instances.

LNL is challenging due to DNNs' tendency to memorize complex and possibly incorrect instances, after initially learning simpler patterns [1, 34]. This can lead to memorization of inaccurate labels, compromising model generalization. Mislabeled data can also cause confirmation bias, which arises when models overfit to early-selected instances. Multi-network and co-training [9, 33] address these issues but add computational overhead and complexity. Both challenges highlight the need for robust training methods.

The "small-loss trick" is a commonly utilized tool for filtering out noisy labels by deeming instances with smaller losses as likely clean. While prevalent in DNN approaches (e.g., [9, 13, 17, 25, 33]), this method is not optimal when noisy and clean example distributions overlap significantly. Alternative methods, like measuring prediction uncertainty over time, have been explored as indicators of label corruption [6, 22, 26].

Similarly to batch selection, LNL methods can be broadly categorized into: loss correction and sample selection [28]. Loss correction includes re-weighting or re-labeling. Active Bias [6] re-weights instances based on prediction variance, bridging LNL and batch selection. A method by Ren et al. [24] uses a clean validation set to dynamically assign weights. Re-labeling refines labels from a mix of noisy labels and DNN predictions, as seen in [23, 29, 32], effectively providing data augmentation. While innovative, these methods pose risks like overfitting to noisy labels and add complexity. Filtering out noisy labels, by contrast, provides a simpler approach but sacrifices information from label refinement. Sample selection filters out mislabeled data during training, often using an auxiliary DNN. MentorNet [13] stands out as a pivotal multi-network approach. It supervises a StudentNet by emphasizing "clean" instances and refining the learning trajectory based on feedback. Han et al. [9] proposed a "Co-teaching" paradigm, an alternative, where two DNNs are trained simultaneously and share insights on small-loss instances to diminish errors from noisy labels. "Co-teaching+" [33] tackles the risk of two networks reaching a consensus with an 'update by disagreement' strategy. The "deep abstaining classifier" [30] is a feature-based multi-network approach, which is particularly effective against structured noise. DivideMix [17] has demonstrated the state-of-the-art LNL performance by employing a semi-supervised (SSL) approach. It dynamically segregates training data into clean and noisy sets using a Gaussian Mixture Model (GMM) and the small-loss trick. To avoid confirmation bias, it utilizes co-teaching. The mislabeled instances are stripped of their labels and refined using SSL [4]. Despite their ability to counteract confirmation bias, the multi-network training approaches often come with significant computational overhead.

SELFIE [26] is a hybrid approach, combining both loss correction and sample selection. SELFIE seeks to refurbish labels of unclean samples selectively, based on uncertainty, and leverage them along with clean samples, to further reduce false corrections while fully exploiting the current training data. While methods like SELFIE are computationally efficient and provide increase in performance, we argue that their selection strategies can be made better. In this paper we focus on improving the batch selection methodology, and compare performance to the pure selection method, Active Bias [6], and two bandit approaches — the *Exponential-weight algorithm for Exploration and Exploitation* (Exp3) [2] and *Follow the Perturbed Leader* (FPL) [16, 21]. Details on these algorithms can be found in Sec. 3 and in Appendix, Sec. A.

**Improving sampling efficiency by exploration**  The challenge of balancing exploration and exploitation is inherent in the process of batch selection. This balance is crucial for identifying new instances that can enhance training efficiency and subsequently leverage them for optimal training outcomes. In addition, it is essential to account for the high degree of non-stationarity in neural network training. Specifically, a network can typically be trained on a particular data instance for only a few iterations before it risks overfitting. To address this issue, our approach aims to manage non-stationarity by dynamically adapting weight estimates. This adaptation can be achieved either through periodic reevaluation, which may be computationally expensive, or by employing a discounted moving average.

So far we are aware of only one study that has directly compared various exploration-exploitation strategies in the context of instance selection: its mixed results suggest such strategies depend highly on the researcher's choice of a sample weight metric [20]. Although $\varepsilon$-greedy and UCB bandit methods have demonstrated effective performance in instance selection [6, 20], the Boltzmann exploration strategy has recently gained prominence in this subfield [5, 19]. In particular, the adversarial bandit — Exp3 [2], which uses Boltzmann exploration, is commonly utilized as a baseline in non-stationary environments, and has been shown to be particularly effective in automated curriculum learning [8, 20].

Our work diverges from prior studies focused on choosing instances [6, 27] or tasks [8, 20], and targets batch selection instead. We utilized the FPL strategy, which can be thought of as a natural extension of Exp3 into combinatorial (batch), rather than individual (instance) action selection.

## 3   Methods

**Adversarial multi-armed bandit problem**  A classic baseline approach for non-stationary environments is the adversarial bandit, in particular, the Exp3 algorithm and its variants. In an adversarial $K$-armed bandit problem, at each time step $t \in \{1, 2, ..., T\}$, the player selects an action $a_t \in \{1, 2, ..., K\}$ and then an adversary, with full knowledge of the player's previous actions, assigns

a reward vector $\mathbf{r_t} = (r_{t,1}, r_{t,2}, ..., r_{t,K}) \in [0,1]^K$ across all actions. The player receives a reward $r_{t,a_t}$ corresponding to the selected action $a_t$. There is typically almost no restrictions on how the adversary can choose the reward vectors, as long as the sequence of reward vectors $\mathbf{r_1}, \mathbf{r_2}, ..., \mathbf{r_T}$ is fixed in advance or chosen based on the player's past actions. The player's goal remains to maximize the total collected reward or equivalently, to minimize regret.

**Combinatorial bandits for batch selection**  In order to select a full batch of instances at once we need to utilize the combinatorial bandit paradigm, which considers the joint utility of combinations of "basic arms". Formally, combinatorial bandits can be considered a type of bandit where a subset of arms is selected in a form of a binary vector $\mathbf{a} \in \{0,1\}^d$, and the final reward is derived from either a Hadamard or a dot product of that vector with the reward vector $\mathbf{r}$. In this work we consider only the subset of a pre-specified batch size $m$, s.t. $||\mathbf{a}||_1 = m$, and a semi-bandit reward model (see Appendix, Sec. A). A direct application of Exp3 to the semi-bandit problem would entail monitoring the sequence of estimates for $\binom{K}{m}$ arms, a task that is computationally infeasible. The state-of-the-art approach to semi-bandits is *Follow the Perturbed Leader* (FPL) [10], which mimics Exp3, but estimates probabilities using reward perturbations, rather than storing them directly. FPL was originally introduced by Hannan [10] and Kalai and Vempala [14], with an efficient version operating on a principle of geometric re-asmpling (GR) proposed by Neu and Bartók [21]. In this work, we adapt the FPL algorithm to batch selection.

FPL operates over $n$ rounds, maintaining a vector of weights $w_{t,i}$ for each action $\mathbf{a}_i$ in the action set $\mathcal{A}$. In our case $\mathbf{a}$ is a binary vector, such that setting an $i$-th action $a_i = 1$ corresponds to selecting an $i$-th instance $\mathbf{x}_i$. Each round the algorithm perturbs the weights with noise $\boldsymbol{\rho}_t$ from distribution $Q$, selecting the action $\mathbf{a}_t$ that maximizes the inner product with the perturbed weight vector. While Neu and Bartók [21] used the $\mathrm{Exp}(1)$ distribution for $Q$, recent work by Honda et al. [11] suggests the Fréchet$(2)$ (also known as inverse Weibull) distribution yields optimal regret in adversarial settings. As opposed to the algorithms presented in literature [11, 21] we estimate reward, rather than loss associated with each arm. We have found this adaptation to significantly improve performance in our application, however we acknowledge that while this algorithm remains in line with reward estimation done in Exp3, the original theoretical performance guarantees for combinatorial arm selection may no longer apply.

---

**Algorithm 1:** Follow The Perturbed Leader (Reward-guided)

**Data:** $\mathcal{A}, n, \eta, M, Q$

1 **for** $i = 1$ **to** $d$ **do**
2 $\quad$ $w_{0,i} \leftarrow 0$ // Initialize weight vector
3 **for** $t = 1$ **to** $n$ **do**
4 $\quad$ Sample $\boldsymbol{\rho}_t \sim Q$ // Sample weight perturbations
5 $\quad$ Compute $\mathbf{a_t} = \arg\max_{\mathbf{a} \in \mathcal{A}} \langle \mathbf{a}, \eta \mathbf{w}_{t-1} + \boldsymbol{\rho}_t \rangle$ // Choose combinatorial action
6 $\quad$ $\mathbf{r}_t \sim \nu_{a_t}$ // Draw reward vector from arm $\mathbf{a}_t$ of MAB $\nu$
7 $\quad$ **foreach** $i$ *with* $a_{t,i} = 1$ **do**
$\quad\quad$ // Geometric Re-sampling
8 $\quad\quad$ Sample $\sigma_{t,i} \sim \mathrm{Geometric}(p_{t,i})$
9 $\quad\quad$ $\hat{r}_{t,i} = \min\{M, \sigma_{t,i}\} a_{t,i} r_{t,i}$ // Compute bounded reward estimate
10 $\quad\quad$ $w_{t,i} = w_{t-1,i} + \hat{r}_{t,i}$ // Update weight of chosen action

---

Following action selection, for each $i$ where $a_{t,i} = 1$, the algorithm proceeds with a geometric re-sampling (GR) step. Sampling from the geometric distribution estimates $1/p_i$ and in practice is not done directly, but rather by sampling arms from $Q + \eta \mathbf{w}_{t-1}$ and counting the number of iterations to re-occurrance. $M$ is the cap on sampling size, to trade off computational efficiency with estimation accuracy. The algorithm draws a sample $\sigma_{t,i}$ from the approximated geometric distribution, and computes a bounded reward estimate $\hat{r}_{t,i}$ in the same way as Exp3, as an importance-weighted estimate, by $a_{t,i}\sigma_{t,i}r_{t,i}$.[1]

Finally, the algorithm updates the weight $w_{t,i}$ of the chosen action $a_i$ by adding the reward estimate $\hat{r}_{t,i}$ to the previous weight $w_{t-1,i}$. This process continues for $n$ rounds, enabling the algorithm to

---

[1]$\sigma_{t,i}$ estimates $1/p_i$

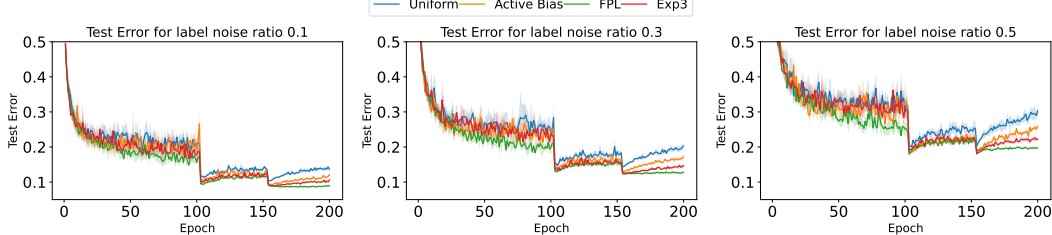

Figure 1: Test error over the course of training with confidence intervals (CI) over 5 runs for Uniform, Active Bias (weighted), Exp3 and FPL, for label noise ratio $\in \{0.1, 0.3, 0.5\}$.

effectively explore and exploit the action space by balancing the current estimated rewards and the exploration noise introduced by perturbations. While FPL may require more computational resources compared to Exp3, it offers the advantage of reducing dependency on the combinatorial action space. This makes FPL a practical choice for real-world sequential decision-making tasks.

**Label noise**    According to Song et al. [28], label noise can be either instance-independent, characterized by constant rates and probabilities, or instance-dependent, where corruption probabilities vary with data features and true labels. This study concentrates on symmetric, instance-independent noise to provide a baseline in a controlled setting.

**LNL weight metric**    Choosing the right metric to select informative instances is still an open problem. In the field of LNL, metrics based on prediction loss [9, 13, 17, 33] and prediction uncertainty [6, 22, 26] have shown particular promise. The following metric was proposed by Chang et al. [6] as part of the Active Bias method:

$$w_i \propto \widehat{\text{var}}(p_{\mathcal{H}_i^{t-1}}(y_i|\mathbf{x}_i)),$$

where, for each instance $\mathbf{x}_i$, it saves prediction probabilities for their target class over time in a history buffer $\mathcal{H}_i$, and then computes their variance.

We employed this metric in our study, as it was shown suitable both for LNL and for batch selection in general. Unlike the metrics that are derived from the change in the state of the model, the probability-based metrics reflect the model's current confidence in its predictions, rendering them independent of the target solutions, and therefore, consistent across instances. This property makes them inherently balanced for problems such as LNL. However, it should be noted that while these metrics offer advantages, they do not directly track the progression of training. Therefore, following Song et al. [27], we limit the size of the history to 10 predictions.

The estimated weights serve two main purposes: either to re-weight the loss as in [24] or to parameterize the probability distribution over data instances. The latter often employs a Boltzmann distribution (e.g. [8, 20]): $P_s(i|H, S_e, D) = e^{w_i/\tau}/Z$ where $Z$ is the normalization constant, $H$ denotes the history of scores (e.g. instance losses or prediciton probabilities), $S_e$ is the set of samples used in the current epoch, and $D = \{(\mathbf{x}_i, y_i) \mid i = 1, 2, \ldots, N\}$ represents the dataset. Given our interest in the role of exploration in sample efficiency, we primarily focus on sampling methods underpinned by bandit algorithms such as Exp3, which also employs a Boltzmann-like distribution.

# 4    Results

**Experimental Setup**    We evaluated the performance of various sampling methods including Uniform Sampling, Active Bias, Exp3, and FPL on the CIFAR-10 dataset using a DenseNet model [12] with 40 layers. We used the Adam optimizer with momentum 0.9 and an initial learning rate of 0.1 that is decayed by multiplicative factor of 0.1 after 40 k and 60 k iterations. The batch size was set to 128 and we ran 200 epochs, consisting of 391 batches each. All methods were repeated 5 times with different seeds, under varying label corruption percentages ranging from 0% to 50%. We report the mean and 95% confidence intervals (CI) of test accuracy achieved by each method. We will release a PyTorch implementation to reproduce our experiments upon paper acceptance.

**Results** Under all label corruption scenarios, FPL exhibited significantly reduced noise and superior performance compared to the other methods, with Exp3 outperforming Active Bias, and Active Bias performing better than Uniform Sampling. In conditions with no label noise, no significant improvement was observed across methods, revealing a potential limitation in sensitivity to "hard to classify" instances and an overfocus on mislabeling.

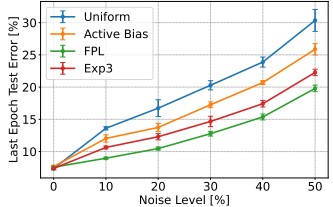

| Noise
Method | | 10% | 20% | 30% | 40% | 50% |
|---|---|---|---|---|---|---|
| Uniform | best | 10.24±0.25 | 12.05±0.54 | 14.08±0.39 | 16.75±0.30 | 19.40±0.26 |
| | last | 13.61±0.24 | 16.73±1.30 | 20.28±0.72 | 23.88±0.78 | 30.34±1.70 |
| Active Bias | best | 9.17±0.15 | 10.69±0.13 | 12.85±0.09 | 15.09±0.19 | 18.09±0.19 |
| | last | 12.03±0.57 | 13.74±0.62 | 17.26±0.45 | 20.70±0.31 | 25.86±0.88 |
| Exp3 | best | 8.74±0.13 | 10.28±0.13 | 12.19±0.15 | 14.51±0.23 | 18.02±0.51 |
| | last | 10.64±0.22 | 12.30±0.49 | 14.67±0.80 | 17.42±0.51 | 22.27±0.50 |
| FPL | best | **8.37±0.19** | **10.04±0.15** | **12.11±0.34** | **14.25±0.36** | **17.65±0.29** |
| | last | 8.97±0.15 | 10.46±0.24 | 12.77±0.37 | 15.37±0.46 | 19.79±0.51 |

Figure 2: Left: Lowest and final epoch test errors (%) for each method on CIFAR-10 by noise ratio. Right: Visualizing last epoch performance.

**Discussion** The methods maintained a consistent ranking across noise levels, with the performance gap widening as noise increased (see Fig. 1 and 2). FPL consistently yielded smooth and stable convergence, due to its ability to choose informative instances. When adopting the same weight metric and neural network architecture as Active Bias, our results show that implementing a bandit strategy can lead to significant performance gains. This underscores the importance of not just selecting an optimal weight metric, but also employing a beneficial exploration policy.

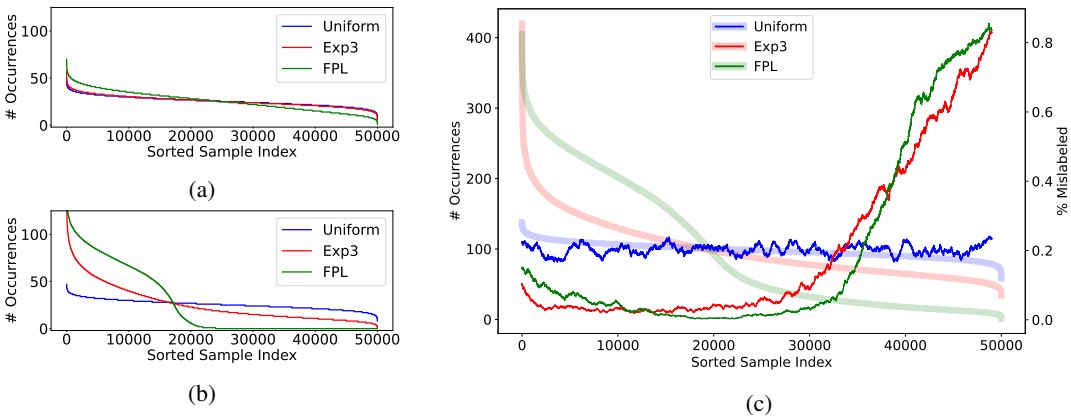

Figure 3: Analysis of instance selection for Uniform, Exp3, and FPL[1] with 20% label noise, showing selection occurrence during the initial (a) and final (b) 1000 iterations and total proportion of mislabeled instances (c; front) over total occurrences (c; background). The order of curves at index 0 aligns well with the overall performance of the methods, revealing a concentration of selection in Exp3 and FPL, particularly pronounced in FPL, with Exp3 demonstrating overfitting to a limited set of instances.

Analysis of instance occurrences (Fig. 3) reveals insights into the differences in sampling strategies, addressing our initial inquiry into performance gain from utilizing batch- as opposed to instance-based feedback. Initially (Fig. 3a), all methods show similar selection frequencies, but distinctions emerge as training concludes (Fig. 3b), especially for Exp3 and FPL. Notably, the algorithm's pattern of concentration on specific instances correlates well with its performance. While Exp3 resembles an exponential distribution, FPL produces a threshold at about 20 k instances, filtering 30 k of the remaining images. The preference for 'clean' instances between the 10 k and 20 k sorted index intensifies towards the end of training, indicating the algorithm's inclination to retain instances

initially deemed 'clean'. These insights emphasize FPL's efficiency as an $m$-set combinatorial bandit method, and highlight its suitability for batch selection.

In Fig. 3c, we display curves representing total counts throughout the run, providing a holistic view of each method's sample selection strategy. Over these, with solid lines, we superimpose the percentage of mislabeled instances within a sliding window of 1000 sorted instances. Each point on the overlay represents the mislabeling percentage within that window, revealing a trend: instances sampled less frequently (toward the right) have higher mislabeling percentages. This visualization supports our hypothesis that bandit methods with uncertainty-based metrics, like Exp3 and FPL, enhance performance by focusing on and filtering out mislabeled instances.

While Exp3 effectively identifies mislabeled instances like FPL (Fig. 3c), it tends to overfit to a narrow set and over-explore the rest (Fig. 3b). This is expected as Exp3 is an instance-based algorithm. However, this overfitting poses risks to its efficacy, as consistently selecting the same subset of impactful instances, combined with a broad array of less pertinent ones, leads to lower performance. Conversely, FPL, by adjusting weights in accordance with other instances, revisits a larger, more balanced subset regularly, forming more informative and consequential batches, ensuring optimal selection in batch training scenarios.

As weights play a crucial role in understanding the learning process in depth, we further analyze the weight dynamics of FPL in Fig. 4. Initially set to 0, the weights adjust smoothly throughout training, maintaining a balance in instance selection without anomalies or overfitting. There is a 40%-60% split in instance selection (Fig. 3b) that is clearly reflected in the weights, with those corresponding to highly informative instances increasing rapidly, whereas those consistently labeled (either correctly or as mislabeled) remaining closer to zero. It is worth noting here that all instances were selected at least once, with all weights turning strictly positive by the end of the run. Entropy visualization (Fig. 4a) further emphasizes effective convergence on a well-sized subset of instances, reinforcing selection for better exploitation without excessive exploration.

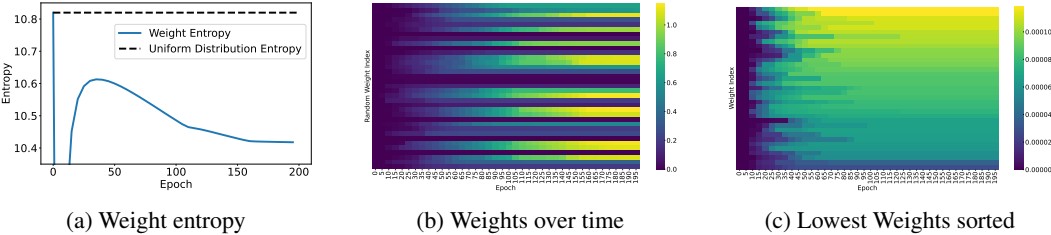

(a) Weight entropy        (b) Weights over time        (c) Lowest Weights sorted

Figure 4: Instance weight visualization. Entropy-aggregated weights are visualized over time in (a). Once all instances are selected by epoch 40, the entropy gradually declines as certain instances gain importance. This pattern indicates effective diversification without overfitting. In (b), a random subset of weights is displayed over time to further validate their individual trajectories. In (c) the lowest weights are shown to dynamics of weights that remain close to 0.

**Scalability and hyperparameter sensitivity**    We ran our experiments using $\eta = 0.3$ and $\gamma = 0.1$ for Exp3 and $\eta \approx 18$, $\beta \approx 20$, and the Fréchet$(0.45)$ distribution for FPL. To show that FPL has small sensitivity to these hyperparameters, we ran a grid search in their vicinity (see Fig. 5). We found the number of GR samples to be optimal between 500 and 1000. In that range GR introduces an additional computational overhead of 20%-40%. This may seem alarming at first, however, we point out that GR is *embarrassingly parallelizable* and instance-based, which makes it scalable in practical applications.

**Limitations and future directions**    Exp3's slower adaptation and potential benefits of its variants like Exp3.P or Exp3.IX warrant further consideration. FPL excels in balancing exploration and exploitation but shows limited improvement in noise-free scenarios, suggesting a potential overfocus on mislabeled instances. While all methods generalize well, tests were limited in scope. Future work will include naturally noisy sets like WebVision [18], as well as metrics like area under the margin (AUM) [22], to deepen insights and enhance results.

---

[1]Active bias method is excluded here as we use its loss re-weighting variant, and so its sampling distribution is the same as uniform.

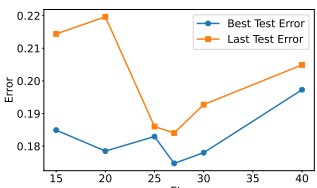 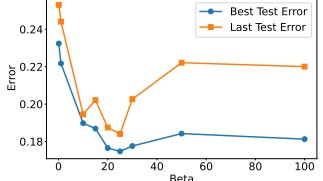 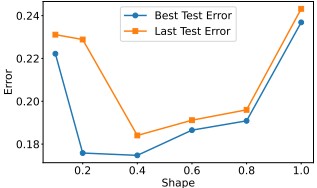

(a) Error w.r.t. Eta (Beta=25.0, Shape=0.4)

(b) Error w.r.t. Beta (Eta=27.0, Shape=0.4)

(c) Error w.r.t. Shape (Eta=27.0, Beta=25.0)

Figure 5: FPL hyperparameter sensitivity analysis for 50% label noise.

## 5   Conclusions

This investigation into the performance of sampling methods under different noise conditions has revealed key insights into their adaptability, stability, and algorithmic nuances. FPL's effective balance between exploration and exploitation, particularly its focus on uncertain instances, underscores its superior performance. Nonetheless, the absence of marked improvement in noise-free settings and the limited scope of our experiments highlight avenues for future research and refinement.

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

## A   Bandit algorithms

**Classic adversarial bandits**   The classic adversarial algorithm, Exp3 (Alg. 8), employs a multi-step process for importance adjustments of reward estimates. First, it adjusts the reward $r_j$ for arm $j$ at time $t$ using the formula $\hat{r}_j = \frac{r_{t,j}}{p_{t,j}}\mathbb{I}_{j=i_t}$, where $p_{t,j}$ is the probability of choosing arm $j$. These adjusted rewards are then used to estimate sample weights $w_j$. Finally, these weights $w_j$ are utilized in the Boltzmann distribution for sampling instances.

---

**Algorithm 2:** Exp3 Algorithm

---

**Data:** $\gamma \in (0,1]$, $K$
1 **for** $i = 1$ **to** $K$ **do**
2     $w_{1,i} \leftarrow 1$ // Initialize weights
3 **for** *each round* $t = 1, 2, ...$ **do**
4     $p_{t,j} \leftarrow (1 - \gamma)\frac{w_{t,j}}{\sum_{k=1}^{K} w_{t,k}} + \frac{\gamma}{K}$ // Compute pmf
5     $i_t \sim p_t$ // Sample action
6     $r_{t,i_t} \sim \nu_{i_t}$ // Draw reward from arm $i_t$ of MAB $\nu$
7     $\hat{r}_j \leftarrow \frac{r_{t,j}}{p_{t,j}}\mathbb{I}_{\{j=i_t\}}$ // Compute reward estimate
8     $w_{t+1,j} \leftarrow w_{t,j}\exp(\gamma\hat{r}_{t,j})$ // Update weights

---

The Exp3 algorithm is very efficient computationally and is suitable for task or individual instance selection, but it doesn't take into account an impact of a full batch of instances on the performance

of the neural network, which may result in suboptimal performance when competing instances are present in the same batch.

**Analysis and adoption of semi-bandit feedback in combinatorial bandits**   Combinatorial bandits can be categorized as full-information, semi-bandits, or full-bandit feedback. In the full-bandit feedback scenario we observe just one reward per batch. While maximizing this reward is our ultimate objective, ignoring the available information about rewards received for individual arms makes the decision process suboptimal. The full information setup, where rewards from all the arms are observed is computationally infeasible, as it requires re-evaluating the network on all the instances. In our research we adopt the semi-bandit, in which the rewards are observed only for the basic arms selected in the current round. This setup aligns well with our problem, where we observe the rewards for instances that the neural network was trained on in current iteration. As this information is readily available, no additional passes through the network are required.

Efficient variant of FPL was proposed by Neu and Bartók [21], who deployed Geometric Re-sampling to estimate probabilities for importance-weighted reward estimates (re-weighting step), making FPL the first computationally feasible solution to semi-bandits with strong guarantees. While other methods may offer comparable or superior theoretical performance in terms of upper regret bounds, they frequently suffer from computational inefficiency or require additional optimization steps, rendering them impractical for real-world applications. The primary appeal of FPL lies in its computational efficiency.

