# OpenReview forum: "Bandit-Driven Batch Selection for Robust Learning under Label Noise"
_NeurIPS.cc/2023/Workshop/WANT — WANT@NeurIPS 2023 Poster_

### Official Review · Reviewer_EXxz · 2023-10-24
**Interesting paper**

**Confidence:** 2

**Review:**

This paper introduces an approach to batch selection, by leveraging combinatorial bandit algorithms and uses it to mitigate the effects of label noise on training deep neural networks.

I found that the paper was well-written and was easy to read. Also, the empirical evaluation seems to indicate that the proposed approach results in tangible benefits in test error for varying levels of label noise. Efficiently and correctly tackling label noise is an open problem with many real-world applications and as such the proposed approach could have a significant impact.

At the same time, the proposed algorithm doesn't seem novel, but its application to the batch selection task is novel (the paper could be a bit clearer on this topic). The empirical evaluation is also quite limited, with only a single architecture and dataset evaluated.

Overall I would recommend that this paper be accepted.

---

### Official Review · Reviewer_VRFY · 2023-10-26
**Interesting work, but with limited experimental support**

**Confidence:** 3

**Review:**

The work presents a method to select training data in the noisy label setting (uniform, label independent noise). The authors propose a new algorithm based on the Multi Armed Bandit (MAB) theory and Learning with Noisy Labels (LNL) methods. Specifically, the proposed method unifies the MAB method called "Follow the Perturbed Leader" (FPL) and a metric first used in the Active Bias method. The idea of the metric is to select arms based on the variance over a sliding window of historical estimates of arms' probabilities.
The authors compare their method with another MAB method called Exp3, as well as with Active Bias on the CIFAR-10 dataset. The results show that in noisy label setting FPL has clear advantage over Active Bias and Exp3 in terms of the final accuracy, and the advantage of FPL grows with noise level. In the noiseless regime FPL has no advantage over other methods.
Although the results of the work are interesting, there are few points related to the article itself and to the data presented, which need to be addressed. Specifically:
1. It is not clear from the text, how exactly the LNL metric is calculated (equation on p. 5, top) and whether this metric was used in the final algorithm.
2. The authors claim that the choice of the reward metric is crucial, but did not provide experiments with alternative choices. Only one metric, namely the one from the Active Bias method, is mentioned. Was the same metric used with Exp3 algorithm as well? This is especially important since the performance of both Exp3 and FPL algorithms is equal in the absence of label noise.
3. Only one dataset (CIFAR-10) was explored. At least one set of experiments on a dataset with natural label noise would be beneficial to strengthen the claims of the work.
4. The advantage of FPL over Exp3 is explained by overfitting of the trained network to a subset of frequently selected examples by Exp3.
It is not clear why this happens. The authors should provide a more in-depth explanation, or to clearly state that this is a hypothesis.

Overall, I rate the research as "Weak accept", since it presents a novel algorithm and achieves a non-trivial improvement over the baselines. However, a firm experimental justification of the approach is missing

---

### Official Review · Reviewer_i8UR · 2023-10-26
**This paper proposes novel combinatorial bandit algorithms for batch selection**

**Confidence:** 4

**Review:**

**Summary of the work**
Overall, I think this is a decent paper for WANT!

This paper studies batch selection in Stochastic Gradient Descent training, leveraging combinatorial bandit algorithms. The proposed method focuses on optimizing the learning process in the presence of label noise, which is a common issue in real-world datasets.

**Strength**

The studied problem is important, and the proposed algorithm is novel and efficient for noisy label setting. Specially, the paper proposes a combinatorial bandit algorithms in curriculum learning setting to select data points in a batch for backward propagation.

The performance improvement does not sacrifice computational overhead commonly associated with neural network models.

**Weakness**

The experiment does not show remarkable improvement in noise-free settings and  the limited scope of the experiments suggest the needs for future research and improvement of the algorithms.

The paper lacks discussions of related paper. For example, https://arxiv.org/pdf/2104.13114.pdf also formulates the data distillation problem (this paper calls it data subsampling) as bilevel optimization problem. It also proposes efficient approximation algorithm to solve the intractable non-convex problem. I'd love to see the authors discussed more about these papers.

For another example, https://arxiv.org/pdf/2306.10728.pdf also proposes similar ideas through automated curriculum learning (CL), which rewards the selection of temporally rare samples In RL/Bandit scenarios.

I would also like to see more analysis in terms of the regret.

---

### Meta-Review · Area_Chair_2i1K · 2023-10-26

**Recommendation:** Accept (Poster)
**Confidence:** 3

**Metareview:**

The authors propose to use combinatorial Bandit algorithms to select data batches, making the data sampling scheme focus on uncertain instances. The reviewer is positive about the paper even if they find the experiments section limited.

---

### Decision · Program_Chairs · 2023-10-28

**Decision:**

Accept (Poster)

**Comment:**

We thank the authors for their time and contribution to WANT and we are pleased to share that after the reviewing process the paper has been accepted. Congratulations! We encourage the authors to consider reviewers' feedback for the improvement of the camera-ready version. We hope to see you in person at the workshop and brainstorm on efficient training research together!